

# The vacua of dipolar cavity quantum electrodynamics

**Michael Schuler**[1*]**, Daniele De Bernardis**[1]**, Andreas M. Läuchli**[2] **and Peter Rabl**[1]

**1** Vienna Center for Quantum Science and Technology,
Atominstitut, TU Wien, 1040 Wien, Austria
**2** Institut für Theoretische Physik, Universität Innsbruck, 6020 Innsbruck, Austria

★ michael.schuler@tuwien.ac.at

## Abstract

The structure of solids and their phases is mainly determined by static Coulomb forces while the coupling of charges to the dynamical, i.e., quantized degrees of freedom of the electromagnetic field plays only a secondary role. Recently, it has been speculated that this general rule can be overcome in the context of cavity quantum electrodynamics (QED), where the coupling of dipoles to a single field mode can be dramatically enhanced. Here we present a first exact analysis of the ground states of a dipolar cavity QED system in the non-perturbative coupling regime, where electrostatic and dynamical interactions play an equally important role. Specifically, we show how strong and long-range vacuum fluctuations modify the states of dipolar matter and induce novel phases with unusual properties. Beyond a purely fundamental interest, these general mechanisms can be important for potential applications, ranging from cavity-assisted chemistry to quantum technologies based on ultrastrongly coupled circuit QED systems.



## Contents



# 1  Introduction

In QED, the relevant dimensionless coupling parameter is the finestructure constant $\alpha_{\rm fs} = E_{\rm C}/E_{\rm ph}$. It can be expressed as the ratio between the Coulomb energy, $E_{\rm C} = e^2/(4\pi\epsilon_0 d)$, of two electrons at a distance $d$ and the energy $E_{\rm ph} = c\hbar/d$, which is needed to create a photon confined approximately to the same region in space. The small value of $\alpha_{\rm fs} \simeq 1/137$ already suggests that the quantized modes of the electromagnetic field play a minor role in the physics of atoms, molecules and solids, as confirmed by more rigorous calculations. However, this argument does not necessarily hold in structured electromagnetic environments, such as nanoplasmonic systems or $LC$ circuits, where the energy of a photon can be tuned independently of its wavelength. In this case the coupling between an electric dipole and a quantized field mode is characterized by an effective parameter $\alpha = \alpha_{\rm fs}(Z/Z_0)$ [1,2], which can be considerably enhanced by increasing the impedance of the mode, $Z$, compared to the value in free space, $Z_0$. This raises an important fundamental question: Can the properties of matter be influenced by such an artificially boosted coupling to the quantized field, and, if so, how would the properties change?

In view of a growing number of experimental setups where $\alpha \sim O(1)$ can potentially be reached [3–6], this question has lately gained additional relevance and first observations of cavity-induced modifications of chemical reactions [7], phase transition points [8,9] and electric transport [10] have been reported. Moreover, the regime $\alpha \gtrsim 1$ [11–13] is already accessible in circuit QED, where artificial atoms in form of superconducting two-level systems are coupled to microwave resonators [14,15]. Already now, such systems offer many intriguing possibilities for investigating basic principles of light-matter interactions in unprecedented coupling regimes. However, due to the complexity of the problem, our understanding of strong-coupling induced modifications of real and artificial matter is still rather poor, even on a conceptual level. In particular, while detailed analytical and numerical computations have already been performed for single molecules in cavities [16–20] or small circuit QED systems [21–24], equivalent calculations are no longer feasible for real materials, strongly correlated electronic systems or larger superconducting devices. The analysis of such systems has thus been constrained to idealized collective-spin models [2,25–30] or to moderate coupling regimes ($\alpha < 1$) [31–41], where in realistic models no significant modifications of ground and thermal states are expected yet [2,16,43–47]. Consequently, still little is known about few- and many-body effects that arise from the direct competition between short-range electrostatic interactions and a non-perturbative coupling to an extended dynamical mode.

In the following analysis we address this open theoretical problem by considering the conceptually simplest scenario of a lattice of interacting two-level dipoles coupled to a single cavity mode. While this prototypical cavity QED system has been the primary workhorse for modelling light-matter interactions in quantum optics and solid-state physics for many decades, its ground state properties are, surprisingly, still unknown. Here we use exact numerical calcula-

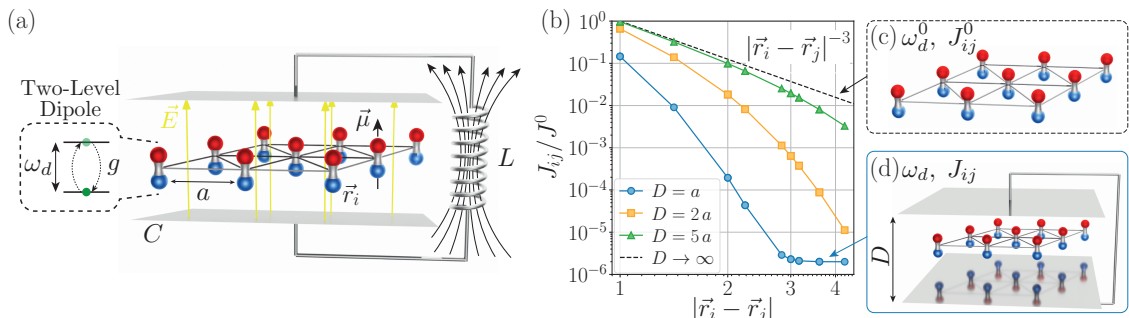

Figure 1: Setup. (a) $N$ anharmonic dipoles with transition dipole moment $\vec{\mu}$, arranged in a two-dimensional lattice with coordinates $\vec{r}_i \perp \vec{\mu}$ and lattice constant $a$, are coupled to a single mode of an $LC$ resonator with frequency $\omega_c = 1/\sqrt{LC}$ and impedance $Z = \sqrt{L/C} \gg Z_0$. The anharmonic dipoles are approximated as two-level systems with transition frequency $\omega_d$. In such a setup, the coupling parameter $g/\omega_c = \vec{\mu} \cdot \vec{E}_0/(\hbar \omega_c) \propto \sqrt{\alpha} \gtrsim 1$ between a single dipole and a single photon, where $\vec{E}_0$ is the electric field per photon, can reach values of order unity. (b) The dipole-dipole interactions $J_{ij}$ depend on the distance $D$ between the metallic plates because of screening effects [see (d)], and differ from the bare values $J_{ij}^0 = J^0/\left|\vec{r}_i - \vec{r}_j\right|^3$ of the dipole system in free space [see (c)]. The screening effects become more important for small $D$ where they strongly reduce the magnitude and range of $J_{ij}$.

tions to evaluate, without any approximations, the non-perturbative effect of vacuum fluctuations on the ground states of strongly-correlated dipolar systems. This analysis confirms, first of all, the existence of so-called superradiant [25–27] and subradiant [2, 23] phases, which already appear in the analysis of simple collective spin models. However, we also observe the formation of completely new phases of dipolar matter and cavity-induced ordering mechanisms, which have not been discussed in the literature before. By that, we are able to provide a first complete phase diagram for the 'vacua' of dipolar cavity QED in elementary lattice geometries. This study also reveals that there is still a wealth of unexplored phenomena in cavity and circuit QED, which may soon become accessible with further experimental advances in these fields.

## 2 Cavity QED of interacting dipoles

We consider a prototypical cavity-QED system as depicted in Fig. 1(a), where $N$ anharmonic dipoles are coupled to a single quantized mode of an $LC$ resonator [29] with frequency $\omega_c$. We approximate the dipoles by two-level systems with transition frequency $\omega_d$, located at fixed lattice positions $\vec{r}_i$ (in units of the lattice spacing $a$). The dipoles couple to the electric field $\vec{E}$ of the cavity with a transition dipole moment $\vec{\mu} \parallel \vec{E}$ and among each other via static dipole-dipole interactions. Under these assumptions the quantized dynamics of the system is described by the Hamiltonian [2] ($\hbar = 1$)

$$H_{\text{cQED}} = \omega_c a^\dagger a + g\left(a^\dagger + a\right)S_x + \frac{g^2}{\omega_c}S_x^2 + \omega_d S_z + \sum_{i<j}\frac{J_{ij}}{4}\sigma_x^i \sigma_x^j, \qquad (1)$$

where $\sigma_\alpha^i$ denote the Pauli operators at site $i$, $S_\alpha = \sum_i \sigma_\alpha^i/2$ are collective spin operators, and $a$ is the annihilation operator for the field mode. Note that $H_{\text{cQED}}$ represents light-matter interactions in the dipole gauge, where gauge-dependent artefacts in the two-level truncation can be avoided [48]. For other recent contributions on this topic, see [49–52].

The cavity affects the dynamics of the dipoles in two different ways. First, in Eq. (1) the dipole frequency, $\omega_d$, and the dipole-dipole couplings, $J_{ij}$, already include screening effects from the metallic boundaries and can differ considerably from their bare values $\omega_d^0$ and $J_{ij}^0$ in free space. This behaviour is illustrated in Fig. 1(b-d), which shows that the usual dipole-dipole interactions, $J_{ij}^0 = J^0/|\vec{r}_i - \vec{r}_j|^3$, become short-ranged and suppressed as the distance $D$ between the plates is decreased. This boundary effect can strongly modify the properties of para- and ferroelectric systems [2, 53–56], but it is of electrostatic origin and not the main focus of this study. Therefore, we simply treat $\omega_d$ and $J_{ij}$ as arbitrary model parameters and investigate the additional modifications induced by the coupling to the dynamical field mode with frequency $\omega_c \sim \omega_d$. For a sufficiently homogeneous mode, these effects are described by the collective dipole-field coupling $g\left(a^\dagger + a\right)S_x$, with a single-dipole coupling strength $g$, and the associated depolarization shift $\sim S_x^2$.

We emphasize that the Hamiltonian $H_{\text{cQED}}$ represents a minimal consistent extension of the more commonly used Dicke model in quantum optics. In particular, the inclusion of the so-called $P^2$-term $\sim S_x^2$ [2, 17, 23, 29, 57] ensures that for $\omega_d \to 0$ we recover the correct electrostatic limit, $H_{\text{cQED}} \simeq \omega_c a^\dagger a + \sum_{i<j} \frac{J_{ij}}{4} \sigma_x^i \sigma_x^j$ [see Appendix B], which is not the case in the aforementioned Dicke model or variants thereof [32, 33, 41]. Therefore, although being based on several simplifying assumptions, $H_{\text{cQED}}$ still respects all Maxwell's equations and allows us to treat electrostatic and electrodynamical interactions in a consistent and transparent manner. For a more detailed derivation and justification of this model in both cavity and circuit QED systems see Refs. [2, 23].

# 3 The ground states of cavity QED

The physics of $H_{\text{cQED}}$ and variants thereof has been studied extensively in quantum optics and solid-state physics, but primarily in the regime $g/\omega_c \ll 1$. In this limit, the system may still feature huge collective Rabi-splittings of $\Omega_R = g\sqrt{N} \sim \omega_c$ in the excitation spectrum [3, 5, 6], but qualitative changes in the ground and equilibrium states are still only perturbative [2, 16, 45–47]. In turn, preceding studies of $H_{\text{cQED}}$ in the non-perturbative coupling regime, $g/\omega_c \gtrsim 1$, have been restricted to very few spins or the special case of all-to-all interactions $J_{ij} = J$ [2, 23, 47]. This strongly reduces the computational complexity of the problem, but also ignores all non-collective correlations, the influence of the lattice geometry and other essential effects. Here we perform exact, large-scale numerical simulations of finite-sized dipole systems to obtain the ground states of $H_{\text{cQED}}$ without any further approximations [see Appendix A for details].

**Phase diagram.** In Fig. 2(a) we first show the ground state phase diagram for $N = 26$ dipoles on a square lattice with nearest-neighbor only couplings $J_{ij} = J\delta_{\langle i,j\rangle}$ and $\omega_d = \omega_c$. For $g = 0$ the cavity is completely decoupled and Eq. (1) reduces to the familiar transverse field Ising (TFI) model. In this limit we observe the expected transition from a disordered paraelectric to the ordered ferroelectric or antiferroelectric Néel phase when $|J|$ exceeds a critical value of $|J^*| \approx 0.7\omega_d$, which agrees within a few percent with the transition point of the infinite system [58].

Although on finite-size systems symmetries cannot be broken spontaneously and the order parameters are strictly zero, the ordered phases can be uniquely identified through the correlations $\langle \sigma_x^i \sigma_x^j \rangle$ between the spins at positions $i$ and $j$. For that, we define the (normalized) structure factor

$$\Sigma_x(\vec{k}) = \frac{1}{N} \sum_{i=0}^{N-1} e^{-i\vec{k}\cdot\vec{r}_{i0}} \langle \sigma_x^i \sigma_x^0 \rangle \tag{2}$$

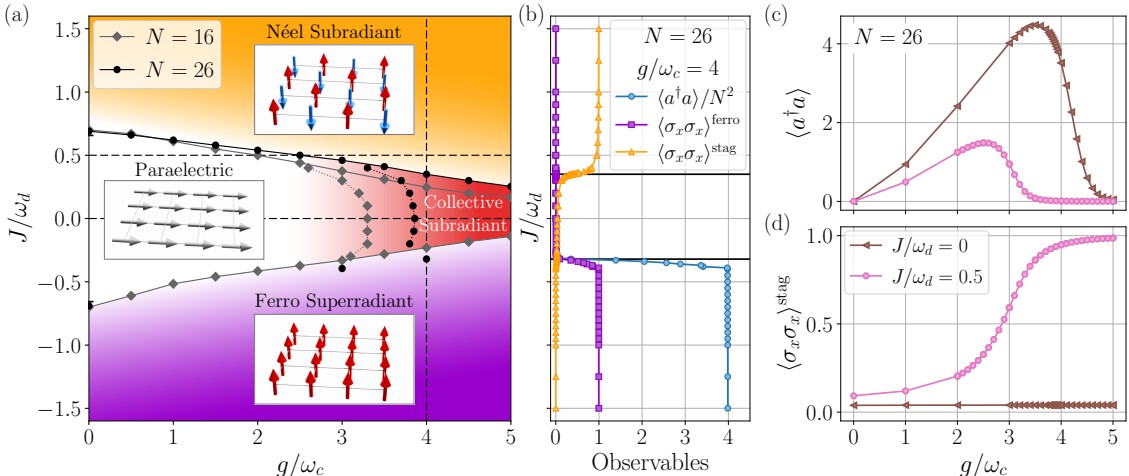

Figure 2: Cavity QED ground states on the square lattice. (a) Phase diagram of $H_{\mathrm{cQED}}$. The symbols track the boundaries between distinct ground states estimated for different system sizes $N$, solid (dotted) lines indicate phase transitions (crossovers). The insets show sketches of the spin configurations in the corresponding phases. (b) Order parameter correlations for a cut at $g/\omega_c = 4$. (c-d) Ground state photon number (c) and order parameter correlations (d) for the transitions from the paraelectric to the subradiant phases with $\langle a^\dagger a \rangle \simeq 0$ along the horizontal dashed lines in (a). For small dipole-dipole interactions, $J/\omega_d$, the collective subradiant regime is stabilized by increasing $g/\omega_c$ while the spins remain disordered (brown curves). For non-zero antiferroelectric interactions, $J/\omega_d > 0$, the system undergoes a phase transition to the Néel subradiant phase, where $\langle a^\dagger a \rangle$ vanishes simultaneously with the onset of antiferroelectric Néel order, $\langle \sigma_x \sigma_x \rangle^{\mathrm{stag}} \to 1$ (pink curves).

for a momentum $\vec{k}$ in the Brillouin zone of the lattice, and $\vec{r}_{i0} = \vec{r}_i - \vec{r}_0$. In an ordered phase $\Sigma_x(\vec{k})$ shows single peaks at specific momenta $\vec{k}^*$, which identify the ordering pattern, and the value of $\Sigma_x(\vec{k}^*)$ can be used to define the ordering strength. To identify the ferroelectric and staggered Néel ordered phases, we introduce

$$
\begin{aligned}
\langle \sigma_x \sigma_x \rangle^{\mathrm{ferro}} &= \Sigma_x(\Gamma), & \text{with } \Gamma = (0,0), \\
\langle \sigma_x \sigma_x \rangle^{\mathrm{stag}} &= \Sigma_x(M), & \text{with } M = (\pi, \pi),
\end{aligned}
\tag{3}
$$

which are nonzero in the corresponding phases, but vanishingly small elsewhere, as shown in Fig. 2(b). These correlation measures are related to the second moments of the order parameters of the respective phases, in particular

$$
\begin{aligned}
\langle \sigma_x \sigma_x \rangle^{\mathrm{ferro}} &= \frac{4}{N^2} \langle p^2 \rangle, \\
\langle \sigma_x \sigma_x \rangle^{\mathrm{stag}} &= \frac{4}{N^2} \left\langle \left(p^{\mathrm{stag}}\right)^2 \right\rangle,
\end{aligned}
\tag{4}
$$

where we have introduced the standard order parameters for the ferroelectric and Néel phases, respectively,

$$
\begin{aligned}
p &\equiv \langle S_x \rangle = \sum_i \sigma_x^i / 2, \\
p^{\mathrm{stag}} &= p_A - p_B = \sum_{i \in A} \sigma_x^i / 2 - \sum_{i \in B} \sigma_x^i / 2.
\end{aligned}
\tag{5}
$$

Here, we have decomposed the lattice into two sublattices $A$ and $B$ to capture the staggered structure of the Néel phase and introduced the sublattice polarizations $p_I = \sum_{i \in I} \sigma_i^x / 2$.

For finite $g/\omega_c \lesssim 1$ the phases do not change considerably, except that in the ferroelectric state now also the photon number acquires a large expectation value, $\langle a^\dagger a \rangle \simeq (gN/(2\omega_c))^2$. Eventually, in the thermodynamic limit $N \to \infty$ the $\mathbb{Z}_2$ symmetry of the model [see Appendix A] is spontaneously broken in the ferroelectric regime leading to a finite spin order parameter $\langle S_x \rangle \neq 0$ and a coherent photonic state with amplitude $\alpha \equiv \langle a \rangle = g/\omega_c \langle S_x \rangle$ [see also Appendix B]. In the quantum optics literature one commonly refers to such a phase as 'superradiant' [25–27], a notation that we adopt in the following. The anti-aligned Néel phase, on the other hand, shows a staggered arrangement of dipoles with $\langle S_x \rangle = 0$ in the thermodynamic limit. While such a state breaks the $\mathbb{Z}_2$ symmetry (in a non-trivial way together with lattice symmetries), the photon sector obeys $\langle a \rangle = 0$, such that we refer to it as 'subradiant'. On finite-size systems, this leads to a strongly reduced photon number $\langle a^\dagger a \rangle \simeq 0$ compared to the fluctuating paraelectric phase [see Fig. 2(c)].

**Collective subradiant phase.** The paraelectric phase gradually evolves into a new 'collective subradiant' phase with unusual properties for $g/\omega_c \gtrsim 3$. This phase exhibits no order and $\langle \sigma_x \sigma_x \rangle^{\text{ferro}} \simeq \langle \sigma_x \sigma_x \rangle^{\text{stag}} \simeq 0$ [see Fig. 2(b)]. At the same time, also the photon number $\langle a^\dagger a \rangle$ vanishes, indicating that all the dipoles are still anti-aligned.

These seemingly contradicting properties can be understood by looking at the limit $J \approx 0$ and $g/\omega_c \gg 1$, where we can eliminate the photons using strong-coupling perturbation theory. The remaining low-energy physics of $H_{\text{cQED}}$ is then approximately described by the effective spin model [2, 23]

$$H_S = \sum_{i<j} \frac{J_{ij}}{4} \sigma_x^i \sigma_x^j + h_z S_z - J_c \left( \mathbf{S}^2 - S_x^2 \right), \qquad (6)$$

where $\mathbf{S} = (S_x, S_y, S_z)$. This Hamiltonian describes Ising interactions with $J_{ij} = J\delta_{\langle i,j \rangle}$ subject to a renormalized 'transverse field' $h_z = \omega_d \exp\left(-g^2/(2\omega_c^2)\right)$ and with an additional cavity-mediated collective coupling of strength $J_c = \omega_d^2 \omega_c/(2g^2) \geq 0$ [see Appendix B].

In the considered regime where the collective subradiant state is observed the term $\propto J_c$ dominates over the exponentially suppressed $h_z$. Thus, for $J = 0$, the Hamiltonian is minimized by a perfectly anti-aligned state $|\psi_{\text{cs}}\rangle$, which obeys $S_x|\psi_{\text{cs}}\rangle = 0$ and has maximal total spin $S = N/2$. Compared to a Néel-ordered configuration, this highly entangled state represents an equal superposition of all possible combinations with half of the dipoles pointing along $x$ and the other half into the opposite direction, without any spatial order. Surprisingly, this peculiar collective phase survives to a high degree in the presence of competing short-range interactions ($J \neq 0$) and it is separated from both ordered phases by a sharp transition.

Although both the collective and the Néel ordered phases are subradiant, a crucial difference between them is visualized in Fig. 2(c-d). These two plots compare the photon number $\langle a^\dagger a \rangle$ and $\langle \sigma_x \sigma_x \rangle^{\text{stag}}$ for fixed $J/\omega_d$, but increasing $g/\omega_c$. For a value of $J = 0.5\,\omega_d$ the system then directly transitions from the paraelectric into the Néel phase. This is signified by an increase of $\langle \sigma_x \sigma_x \rangle^{\text{stag}}$ and simultaneously, or better to say as a consequence of that, also the photon number vanishes. In the evolution from the paraelectric to the collective subradiant states, as shown exemplarily for $J = 0$, the staggered polarization is always vanishingly small, but the dipoles still completely decouple from the photons for large $g/\omega_c$. The formation of such a collective subradiant phase is thus much more subtle than an interaction induced spatially ordered anti-alignment of dipoles.

**Estimating phase boundaries.** After establishing the different phases which appear as vacua of $H_{\text{cQED}}$ on the square lattice, let us in the following estimate their boundaries in the phase diagram. The ordered phases (ferroelectric and Néel) are separated from the disordered phases

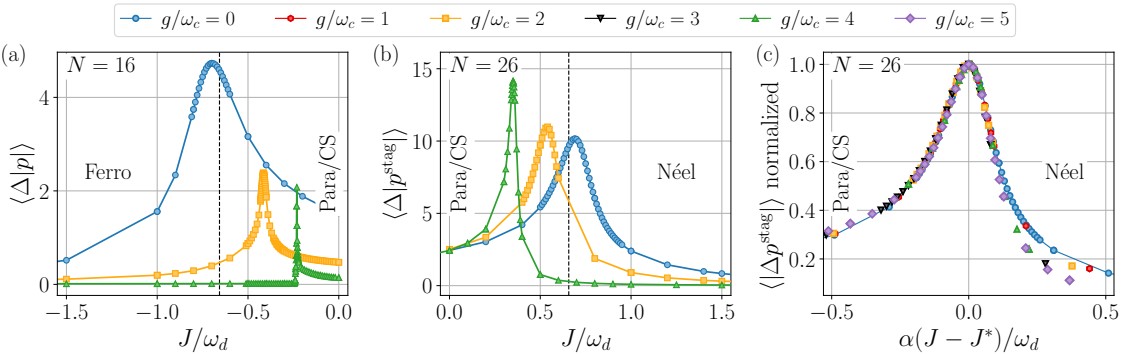

Figure 3: Order parameter fluctuations. Fluctuations for the phase transitions between the disordered and (a) ferroelectric, (b) Néel phases as a function of the dipole-dipole interaction $J/\omega_d$. The peaks are used to estimate the finite-size phase boundaries $J^*(g/\omega_c, N)$ [see also Fig. 2(a)]. The dashed vertical lines indicate the critical point $J^*(g = 0)$ in the thermodynamic limit $N \to \infty$ [58]. With increasing coupling $g/\omega_c$, we observe a cavity-induced reduction of $|J^*|/\omega_d$ and a narrowing of the critical region, which is estimated by the width of the peaks. (c) The fluctuations $\langle |\Delta p^{\mathrm{stag}}| \rangle$ normalized by the height of their peak are plotted against the rescaled distance from the critical point $J^*(g/\omega_c)$, where the rescale factor $\alpha$ depends on $g/\omega_c$. We observe a good collapse with the fluctuations in the transverse field Ising limit $g/\omega_c = 0$.

by a phase transition, where the model's symmetry is spontaneously broken (for $N \to \infty$). The finite-size phase boundaries can be clearly identified by sharp peaks in the order parameter fluctuations

$$\langle \Delta \mathcal{O} \rangle = \langle \mathcal{O}^2 \rangle - \langle \mathcal{O} \rangle^2, \tag{7}$$

as shown in Fig. 3. The finite-size results for $g = 0$ agree within a few percent with the known phase boundaries in the thermodynamic limit. With increasing $g$ we observe a significant cavity-induced reduction of the critical coupling strength $|J^*(g)|$ for both ordered phases. Within the limited range of available system sizes we still see a weak dependence of the phase boundaries on the particle number $N$ [see Fig. 2(a)], but also for larger $N$ no significant qualitative changes of the phase transitions are expected.

Compared to the $g = 0$ results, where the transitions are known to be continuous and in the (2+1)D Ising universality class, with increasing $g/\omega_c$ the width of the peaks in the fluctuations shrinks, which indicates a narrowing of the critical region. Moreover, for the transition between the ferroelectric and disordered phases, the shape of the fluctuations changes when $g/\omega_c \gtrsim 4$, while it remains the same (up to rescaling) for the transition between the Néel and the disordered phases, as shown in Fig. 3(c). Although numerical evidence is still limited, this behavior indicates a cavity-induced change from a continuous to a first order phase transition in the ferroelectric case, while the transition into the Néel phase remains continuous for all $g/\omega_c$.

The evolution from the paraelectric to the collective subradiant regimes, on the other hand, does not show any trend towards a non-analytic behaviour in our analysis. Since also both of these regimes do not break any symmetries, we expect this evolution to be better described by a smooth crossover, instead of a sharp phase transition. As such there is no well-defined transition line and in our plots we use the characteristic peak in the polaron photon number $\langle a^\dagger a \rangle^{\mathrm{polaron}}$ to define a crossover boundary [see Appendix F]. For this crossover we observe a non-negligible dependence on $N$, which in detail depends on the observable that is used to define the boundary. In the non-interacting case $J = 0$ the evolution of the crossover bound-

ary can be numerically estimated also for much larger values of $N$ [see Appendix F]. These simulations further support a smooth crossover between the paraelectric and the subradiant phase, but do not yet provide a conclusive picture about the behaviour of this crossover in the limit $N \to \infty$.

Finally, let us emphasize that $H_{\text{cQED}}$ describes the coupling of all dipoles to a single cavity mode with fixed properties, in particular, a fixed interaction region. Simply increasing $N$ does not represent a well-defined thermodynamic limit, while a rescaling of the coupling constant, $g \to g/\sqrt{N}$, would render the dipole-field coupling non-perturbative. Therefore, the finite-size phase diagrams discussed so far and in the following are already representative for the practically relevant scenarios, where small or mesoscopic ensembles of dipoles are coupled to a field mode localized within a tiny mode volume.

## 4  Order and fluctuations

From the analysis above we can extract two basic cavity-induced many-body effects, namely the stabilization of phases with pre-existing order and the suppression of fluctuations in the disordered phase through the formation of highly entangled collective states. One thus expects that also in general cavity-induced modifications will be most significant in situations where strong fluctuations occur already in the bare system. As a prototypical example we now consider the ground state phases of $H_{\text{cQED}}$ for repulsive dipoles on a triangular lattice [see also Fig. 6], where we again assume nearest-neighbor interactions $J_{ij} = J\delta_{\langle i,j\rangle}$. In this configuration the dipole-dipole interactions are strongly frustrated and lead, for $g = 0$, to large fluctuations (in $S_x$) even in the ordered ground states at $J > 0$. As shown in the corresponding full phase diagram in Fig. 4(a), under this condition completely new cavity-induced phases appear at sufficiently large $g/\omega_c$.

To understand these observations, let us first summarize the established results of the frustrated TFI model at vanishing coupling $g = 0$. In the classical limit $J/\omega_d \to \infty$, the strong frustration prevents the spins from ordering even at zero temperature and the model exhibits an exponentially large (in $N$) ground-state manifold with a finite $T = 0$ entropy density [59]. However, quantum fluctuations from a transverse field, i.e. the term $\omega_d S_z$ in Eq. (1), select an ordered subset of states in an "order-by-disorder" (OBD) process [60]. As shown in the inset in Fig. 4(a), the selected three-sublattice (3SL) antiferroelectric state [61,62] can be depicted as an arrangement of anti-aligned dipoles on two of the sublattices (in the $S_x$ direction), while on the third sublattice the dipoles align (paraelectrically) with the transverse field and do not point in any particular direction according to $S_x$. This phase is thus characterized by a long-range 3SL order, while it still exhibits strong fluctuations in $S_x$ [see Fig. 4(c), left panel]. When the interaction strength $J$ is decreased below a critical value, the 3SL phase eventually becomes unstable towards a disordered, paraelectric phase. The phase transition is continuous and features an emergent $O(2)$ symmetry, such that the universality class is not of Ising, but of XY type [61,62].

To investigate the properties of this system for $g/\omega_c > 0$, we compute the correlations corresponding to 3SL order

$$\langle \sigma_x \sigma_x \rangle^{\text{3SL}} = \Sigma_x(K), \quad \text{with } K = (\pm 4\pi/3, 0), \tag{8}$$

and find an extended region above a critical line $J^*(g)$ where they become large. This indicates the stability of the 3SL phase also in the presence of the cavity mode [see Fig. 4(a-b)]. Similar to the square lattice case, increasing the coupling to the cavity reduces the critical value $J^*(g)$, which we estimate by maxima in the fluctuations $\langle \Delta |p^{\text{3SL}}| \rangle$ for the complex 3SL

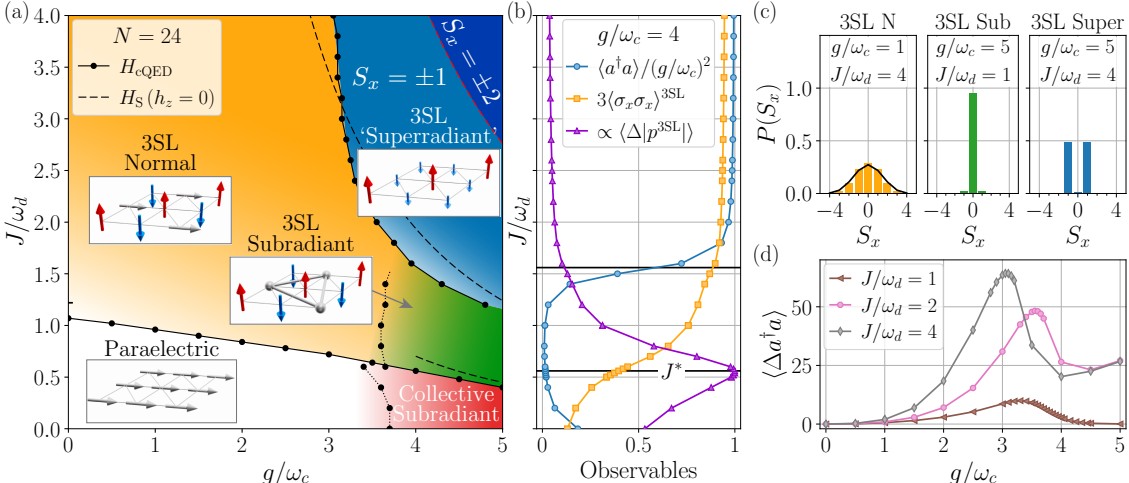

Figure 4: Cavity QED ground states on the triangular lattice. (a) Phase diagram of $H_{\text{cQED}}$ for a system of $N = 24$ dipoles. The insets show illustrations of the spin configurations in the corresponding phases. Solid (dotted) lines indicate phase transitions (crossovers). The dashed lines show phase boundaries estimated from the effective spin model $H_S$ with $h_z = 0$ [Eq. (6)] and show good agreement with the exact phase boundaries from $H_{\text{cQED}}$ for large $g/\omega_c$. The red dashed line indicates a transition between superradiant phases with different values of $S_x$, which appears for larger lattices. The shown boundary is estimated for $H_S$ with $h_z = 0$ and $N = 36$ [see also Fig. 5]. (b) Observables for a cut at $g/\omega_c = 4$. The correlations $\langle \sigma_x \sigma_x \rangle^{\text{3SL}}$ reveal spin ordering above a critical interaction strength $J^*(g)$ which is estimated by a peak in $\langle \Delta | p^{\text{3SL}} | \rangle$. The photon number $\langle a^\dagger a \rangle$ clearly discriminates the transition between the sub- and superradiant 3SL phases. (c) Polarization histograms in the 3SL normal, subradiant, and superradiant phases (from left to right). The solid line in the left panel shows, as a comparison, the expected histogram for a single paraelectric sublattice. (d) Photon number fluctuations. The maxima are used to track the phase boundaries to the 3SL superradiant phase, where the fluctuations show a non-zero value.

order parameter

$$p^{\text{3SL}} = p_A + p_B\, e^{-i4\pi/3} + p_C\, e^{i4\pi/3}, \tag{9}$$

where the lattice was decomposed into the three sublattices $A$, $B$, $C$.

For $J < J^*(g)$ we observe the crossover between the paraelectric and the collective subradiant phase also for the triangular lattice, since the geometry becomes irrelevant in this regime. While the formation of such a homogeneous state is hindered by the 3SL order above the transition line $J > J^*(g)$, we discover a new type of '3SL subradiant' regime for $g/\omega_c \gtrsim 3$. This regime is characterized by a finite order, $\langle \sigma_x \sigma_x \rangle^{\text{3SL}} > 0$, and is thus separated from the collective phase by a sharp transition line [see Fig. 4(a-b)]. At the same time it differs from the normal 3SL phase in terms of its vanishing photon number, $\langle a^\dagger a \rangle \simeq 0$, which indicates strongly reduced polarization fluctuations. This difference can be clearly seen in the ground state distribution of $S_x$-values in the two regimes, as shown in Fig. 4(c): while the polarization distribution is broad in the normal 3SL phase, it is pinned to a single value of $S_x = 0$ deep in the subradiant regime. This behavior can be intuitively explained, by adopting again the simplified picture of a 3SL state, where the fluctuating dipoles on one sublattice participate in the formation of a collective subradiant configuration, similar to the state $|\psi_{\text{cs}}\rangle$, while the two $S_x$-polarized sublattices remain unaffected. Note, however, that this is only an oversim-

plified picture of the actual state, where correlations among different sublattices do not vanish completely.

# 5 Order by cavity-induced disorder

A very surprising finding in the case of a triangular lattice is the appearance of an additional superradiant phase for antiferroelectric dipole interactions [blue region in Fig. 4(a)]. As shown by the histogram in Fig. 4(c), also in this phase the polarization is well-defined, but assumes non-zero values $S_x = \pm 1, 2, \ldots$, and consequently, $\langle a^\dagger a \rangle = (g/\omega_c\, S_x)^2 \gg 1$. Although this value is much smaller than in the regular superradiant phase, this property is completely unexpected, since, at first sight, in this regime both the direct dipole-dipole as well as the cavity-induced interactions would favor a fully anti-aligned configuration. As shown in Fig. 4(d) the transition into this phase is associated with a sharp peak in the photon-number fluctuations, $\langle \Delta a^\dagger a \rangle$, which also remain finite within this phase.

To further investigate the properties of this new type of superradiant states we focus on the relevant regime $g/\omega_c \gg 1$, where, as shown above, we can eliminate the photons using strong-coupling perturbation theory to obtain the effective spin model $H_S$ [Eq. (6)]. Although it is derived under the assumption $J_{ij} \to 0$, a comparison with full numerical simulations up to $N = 24$ shows that $H_S$ accurately reproduces the qualitative features of $H_{\mathrm{cQED}}$ for large $J_{ij}$, as long as $g/\omega_c \gtrsim 3$ [see Fig. 4(a)].

As already discussed above, the regular OBD process on the frustrated, antiferroelectric Ising interactions (for $g/\omega_c \lesssim 3$) is driven by the perturbation with a transverse field $\propto S_z$ [i.e. $H_S$ with artificially set $J_c = 0$], which stabilizes the normal 3SL phase. On top of that, the cavity-mediated collective term $\propto J_c$ can induce a crossover into the 3SL subradiant regime. The appearance of a superradiant phase suggests that this hierarchy no longer holds for large $J/\omega_d$ and $g/\omega_c \gtrsim 3$. In this regime, the transverse field $h_z$ is already strongly (exponentially) suppressed and the cavity-mediated collective term $\propto J_c$ is the dominant perturbation on the Ising interactions. Its quantum fluctuations select a distinctly ordered subset of states compared to the regular OBD process and yield a superradiant phase with a different 3SL ordering pattern.

A common way to analyze different ordering patterns on the triangular lattice is to look at the distribution of the 3SL order parameter

$$p^{\mathrm{3SL}} = \left| p^{\mathrm{3SL}} \right| e^{i\theta} \tag{10}$$

in the complex plane [see also Appendix E]. In Fig. 5(a) we use this method to represent the different ground-state structures in the normal, the subradiant and the superradiant 3SL phases. The large values of $\langle |p^{\mathrm{3SL}}| \rangle$ show that all three phases exhibit 3SL order, as also seen from $\langle \sigma_x \sigma_x \rangle^{\mathrm{3SL}}$ in Fig. 4(b), but with different levels of fluctuations. Even more, the pattern for the superradiant phase differs qualitatively from the other two plots, in particular the positions of the largest peaks are shifted, and indicate a configuration where dipoles in one sublattice are (almost) fully polarized in $S_x$, while dipoles on the other two sublattices are equally, but only partially polarized along the opposite direction. For $N = 24$ this ordering leaves a residual net polarization $S_x = \pm 1$.

Within the effective spin model we can investigate the superradiant phase also for larger lattices and find that this residual polarization increases with the system size and leads, with increasing ratio $J/J_c$, to a whole series of superradiant phases, characterized by $|S_x| = 1, 2, 3, \ldots, S_x^{\max}$ [see Fig. 5(b)]. The maximum value $S_x^{\max}$ obtained in the OBD limit $J_c/J \to 0$ can be calculated from first order degenerate perturbation theory [see Appendix D] and is plotted in Fig. 5(c) for different (regular) triangular clusters of up to $N = 48$ sites.

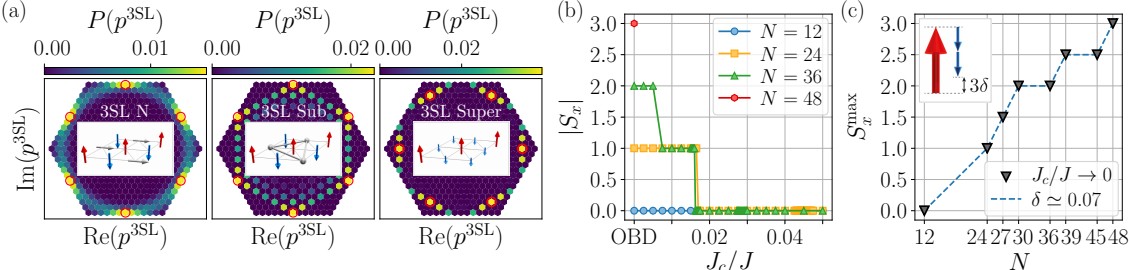

Figure 5: Cavity-induced OBD mechanism in the large coupling limit from the effective spin model. (a) Sublattice polarization distribution of the complex order parameter $p^{3SL}$ for the normal, subradiant, and superradiant 3SL phases (from left to right) for $N = 36$ spins. The large value of $\langle |p^{3SL}| \rangle$ in all panels indicates strong 3SL order for all these phases, while the different positions of the maxima (indicated by the red circles) show that the ordering pattern for the superradiant phase differs from the others. The fluctuations around the maxima identify the strongly distinct nature of the unpolarized sublattice in the normal and subradiant 3SL phases. (b) The ground state $S_x$ sector cascades from a $N$-dependent maximal value $S_x^{max}$ to zero in the 3SL subradiant regime when $J_c/J$ is increased. (c) Maximal ground state polarization $S_x^{max}$, achieved in the OBD limit $J_c/J \to 0$, as a function of system size $N$. The dashed line shows the expected behavior for a polarization density $\delta \simeq 0.07$.

From this analysis we can extract a linear scaling for the maximal ground state polarization $S_x^{max} = \delta \cdot N$ and the photon number $\langle a^\dagger a \rangle = (g\delta \cdot N/\omega_c)^2$ with a polarization density of $\delta \simeq 0.07$. Interestingly, very similar distributions of $p^{3SL}$ and a finite net polarization (although at a much smaller value of $\delta$) have been previously discussed in connection with supersolidity in frustrated spin systems [63–67], where magnetic and superfluid order parameters coexist. While outside the scope of the current study, this connection between superradiance and supersolidity in cavity QED is a particularly exciting direction to explore further. Finally, we want to point out that the interplay between short-range and cavity-mediated collective interaction, such as in $H_S$, has been a source to observe unexpected superradiant and supersolid phases in experiments with cold atoms in cavities [68, 69]. While the interactions and the nature of the phases in these systems are different, interesting connections between the different models might be explored in future works.

## 6 Conclusions

In summary, we have addressed the many-body problem in cavity QED, which arises from the interplay between short-range electrostatic interactions and the non-perturbative coupling to a common cavity mode. Based on exact numerical calculations, we have obtained a first complete phase diagram for the 'vacua' of cavity QED covering the full range of dipole-field interaction strengths, from vanishingly small to ultrastrong. By taking the influence of short-range dipole-dipole interactions in different lattice geometries fully into account, we have shown how the competition between conventional and cavity-induced correlations can lead to the formation of several novel phases, which have no direct counterparts in the collective Dicke-type models [25–27, 30] usually studied in quantum optics, nor in regular solid-state spin systems. Although we focused here on the conceptually simplest case of two-level dipoles, the basic mechanisms identified in this work, i.e. the cavity-induced reduction of fluctuations, extended subradiant states without order, or a cavity-induced OBD process, will also be rele-

vant in various other strongly interacting cavity QED systems [36–38, 40–42], where so far the analysis of ground states has been constrained to mean-field methods or perturbative coupling regimes.

While the most interesting regime of light-matter interactions, $g/\omega_c > 1$, is not accessible in cavity QED experiments with atoms and molecules today, recent advances in the fabrication of electromagnetic resonances with $Z/Z_0 > 50$ [13, 70] show that this regime is by no means out of reach. Of immediate relevance are our findings for the field of circuit QED, where equivalent models can also be obtained with alternative galvanic coupling schemes, and values of $g/\omega_c > 1$ [11, 12] have already been demonstrated for single superconducting two-level systems. The extension of these experiments to larger arrays of superconducting qubits coupled directly and via microwave modes will provide a natural platform to explore the new phases and physical mechanisms identified in this work. Such systems are currently developed for quantum simulation and quantum annealing schemes [71], where ultrastrong coupling effects, similar to what we have analyzed here, can find direct practical applications [72].

# Acknowledgements

We thank Alessandro Toschi for stimulating discussions and feedback on the manuscript.

**Funding information** This work was supported by the Austrian Academy of Sciences (ÖAW) through a DOC Fellowship (D.D.) and by the Austrian Science Fund (FWF) through the DK CoQuS (Grant No. W 1210) and Grant No. P31701 (ULMAC). The computational results presented have been achieved [in part] using the Vienna Scientific Cluster (VSC).

# A  Numerical simulations

The numerical results in this manuscript have been achieved by Exact Diagonalization using a Lanczos algorithm [73, 74] on regular clusters with a finite number of $N$ two-level systems [see Fig. 6]. To reduce finite-size effects we use periodic boundary conditions along both directions of the square and triangular lattices and, to study genuine two-dimensional properties we only use clusters with an aspect ratio $\epsilon = 1$, i.e. the loops around both periodic directions have equal length. To fit the antiferroelectric phases with a two (three) sublattice structure, we only consider square (triangular) clusters with $N \mod 2 = 0$ ($N \mod 3 = 0$). Here, it is worth mentioning that the subradiant states discussed in this work cannot exist on clusters with odd $N$, where the possible total polarizations $S_x$ are half-integers, so that a subradiant state with fixed $S_x = 0$ cannot be obtained.

The Hilbertspace $\mathcal{H}$ is kept finite by additionally introducing a photon-number cutoff $n_{\text{ph}}^{\max}$ for the cavity mode in $H_{\text{cQED}}$, such that $a^\dagger |n_{\text{ph}}^{\max}\rangle \equiv 0$ and $\dim[\mathcal{H}] = 2^N \left(n_{\text{ph}}^{\max} + 1\right)$. $n_{\text{ph}}^{\max}$ has to be chosen large enough to achieve accurate results throughout the different regimes of the external parameters [see Appendix C for a thorough discussion]. It can also be favorable to transform $H_{\text{cQED}}$ into the polaron frame, which describes a distinct basis, where, in some regimes, a much smaller cutoff than in the original frame can be sufficient [see Appendix B].

To further reduce the Hilbertspace dimension, we use the $\mathbb{Z}_2$ symmetry of $H_{\text{cQED}}$, given by the operator $\mathcal{S} = \mathrm{e}^{-\mathrm{i}\pi(a^\dagger a + S_z)}$, together with the lattice translational and point-group symmetries to block-diagonalize $\mathcal{H}$.

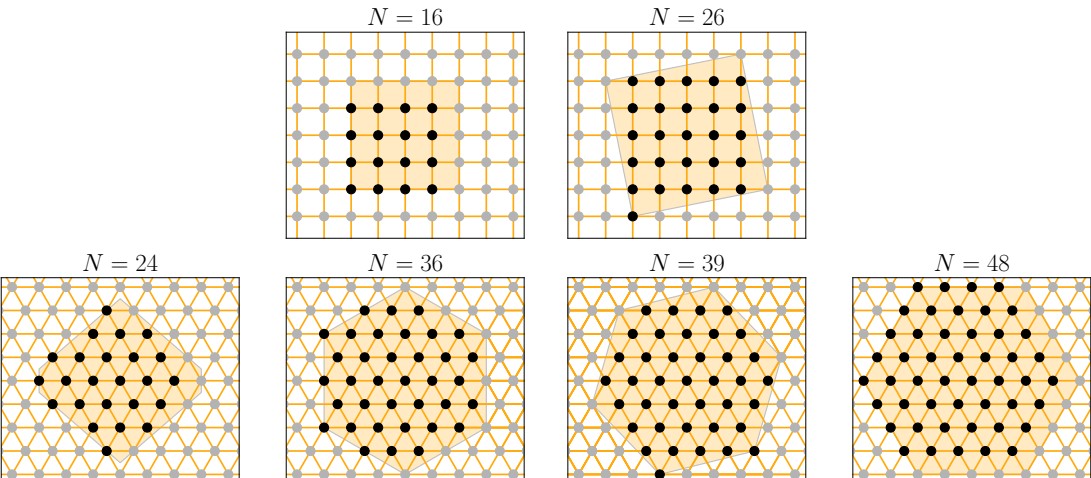

Figure 6: Some of the finite-size clusters used in the simulations. The top (bottom) row shows square (triangular) clusters. The black dots are the sites of the finite-size clusters, the yellow background illustrates the Wigner-Seitz cell. The yellow lines indicate the nearest-neighbor bonds.

## B  Polaron transformation & effective spin model

In the ultrastrong coupling regime it can be convenient to transform $H_{\text{cQED}}$ [Eq. (1)] into a frame of displaced photon number states (polarons) by applying the unitary operator $\mathcal{U} = e^{g/\omega_c S_x(a^\dagger - a)}$. The Hamiltonian $H_{\text{cQED}}$ transforms into

$$
\begin{aligned}
\widetilde{H}_{\text{cQED}} &= \mathcal{U} H_{\text{cQED}} \mathcal{U}^\dagger \\
&= \omega_c\, a^\dagger a + \sum_{i<j} \frac{J_{ij}}{4} \sigma_x^i \sigma_x^j + \omega_d\, \mathcal{U} S_z \mathcal{U}^\dagger,
\end{aligned}
\tag{11}
$$

since $a^{(\dagger)} \to a^{(\dagger)} - g/\omega_c\, S_x$ is displaced proportional to $S_x$. Within this formulation, it becomes obvious that the correct electrostatic limit $H_{\text{cQED}} \simeq \omega_c a^\dagger a + \sum_{i<j} \frac{J_{ij}}{4}\sigma_i^x \sigma_j^x$ is achieved for $\omega_d \to 0$, since the depolarization shift in Eq. (1) exactly cancels the additional terms $\propto S_x^2$ from the transformation of $a^\dagger a$.

The Hamiltonian Eq. (11) can also be advantageous to study superradiant phases, because the photon number $\langle a^\dagger a \rangle^{\text{polaron}}$ in this polaron frame ignores the part from the direct coupling to the polarization $S_x$ and remains much smaller than in the standard frame, in particular for superradiant phases. Therefore, a substantially lower photon number cutoff $n_{\text{ph}}^{\max}$ can be sufficient for precise numerical simulations, with the disadvantage of having to deal with a dense photonic Hamiltonian, when $\omega_d \neq 0$.

Also, the polaron photon number, which can be computed by $\langle a^\dagger a \rangle^{\text{polaron}} = \left\langle (a^\dagger - \alpha)(a - \alpha) \right\rangle$ with $\alpha = g/\omega_c\, S_x$ in the standard frame, can be a useful observable. In particular, we use characteristic peaks in the polaron photon number to identify the crossover regime between the paraelectric and collective subradiant phases [see Appendix F].

Using strong-coupling perturbation theory for $g/\omega_c \gg 1$ and projecting onto the lowest-energy sector without polaronic excitations $|0\rangle_{\text{ph}}^{\text{polaron}}$, the last term in $\widetilde{H}_{\text{cQED}}$ can be approximated as [2, 23]

$$
\omega_d\, \mathcal{U} S_z \mathcal{U}^\dagger \simeq \omega_d e^{-\frac{g^2}{2\omega_c^2}} S_z - \frac{\omega_d^2 \omega_c}{2g^2}\left(\mathbf{S}^2 - S_x^2\right),
\tag{12}
$$

where we have introduced the total spin operator $\mathbf{S} = (S_x, S_y, S_z)$. Within this approximation, we thus obtain the effective spin model $H_S$ given in Eq. (6). It is important to note that the eigenstates in the original basis $|\Psi\rangle = e^{-g/\omega_c S_x(a^\dagger - a)} |\Psi\rangle_{\text{spin}} \otimes |0\rangle_{\text{ph}}^{\text{polaron}}$, and the photon number $\langle a^\dagger a \rangle = g^2/\omega_c^2 \langle |S_x| \rangle^2$ is generally non-zero in the standard basis. Note that the approximate expression in Eq. (12) has been derived for non-interacting dipoles $J_{ij} = 0$.

## C  Photon number cutoff

In this appendix we discuss the implications of introducing a photon number cutoff $n_{\text{ph}}^{\max}$ to obtain a finite Hilbert space. This cutoff has to be chosen large enough such that the true ground state in the full Hilbert space only shows negligible deviations (up to some defined precision) when it is projected into the restricted Hilbert space. Appropriate values for $n_{\text{ph}}^{\max}$ strongly depend on the chosen external parameters, i.e., for parameters belonging to a superradiant phase much larger cutoffs have to be chosen than for parameters belonging to a subradiant phase.

For the simulations in this work we choose $n_{\text{ph}}^{\max}$ large enough, such that doubling this cutoff does not change the measured observables. In Fig. 7 we show an analysis of the dependence of observables on $n_{\text{ph}}^{\max}$ for a square lattice configuration with $N = 16$ dipoles and a constant coupling $g/\omega_c = 2$. For antiferroelectric interactions, $J/\omega_c > 0$, a small cutoff $n_{\text{ph}}^{\max} = 64$ is already sufficient to obtain converged results in all observables, since the average photon number $\langle a^\dagger a \rangle$ remains small. Contrarily, for ferroelectric interactions, $J/\omega_c < 0$, the average photon number and its fluctuations become large [see Fig. 7(a)] when the superradiant regime is entered. Then, a too small cutoff yields false results not only for $\langle a^\dagger a \rangle$, but also for pure dipole observables [see Fig. 7(c)-(f)]. The distribution of the photon number $a^\dagger a$ in the ground state [see inset in Fig. 7(a)] reveals, that a cutoff $n_{\text{ph}}^{\max} \gtrsim 400$ would be sufficient for a simulation with these particular external parameters.

In Fig. 7 we also show results obtained from the polaron frame with Hamiltonian $\widetilde{H}_{\text{cQED}}$. In this formulation, a much smaller cutoff is already enough to obtain converged results in the ferroelectric regime, since the polaron photon number $\langle a^\dagger a \rangle^{\text{polaron}}$ remains small even in the superradiant phase [see Fig. 7(b)].

## D  OBD simulations

The classical Ising model, which is obtained from $H_{\text{cQED}}$ or $H_S$ for $J \to \infty$, is strongly frustrated on the triangular lattice with nearest-neighbor interactions and does not order even at zero temperature $T = 0$. It features an exponentially large (in $N$) ground-state manifold, with an extensive $T = 0$ entropy $S \approx 0.323 k_B N$ [75]. This ground-state manifold can be destabilized by quantum fluctuations from other interaction terms, such as a transverse field or the cavity-mediated collective coupling $\propto J_c$ in $H_S$, when a particular subset of states, with the softest response to the fluctuations, is selected. If the set of the selected states is ordered, this process is termed "order by disorder" [60], and for a perturbation with a transverse field this mechanism is known to induce a 3SL ordered phase [61, 62].

To study the OBD process from the collective coupling $\propto J_c$ in Eq. (6) in the $J/J_c \to \infty$ regime, we restrict the Hilbertspace of the system to the degenerate, classical ground-state manifold, and define the effective Hamiltonian

$$H_{\text{OBD}} = -\mathcal{P}_J \left( \mathbf{S}^2 - S_x^2 \right) \mathcal{P}_J. \tag{13}$$

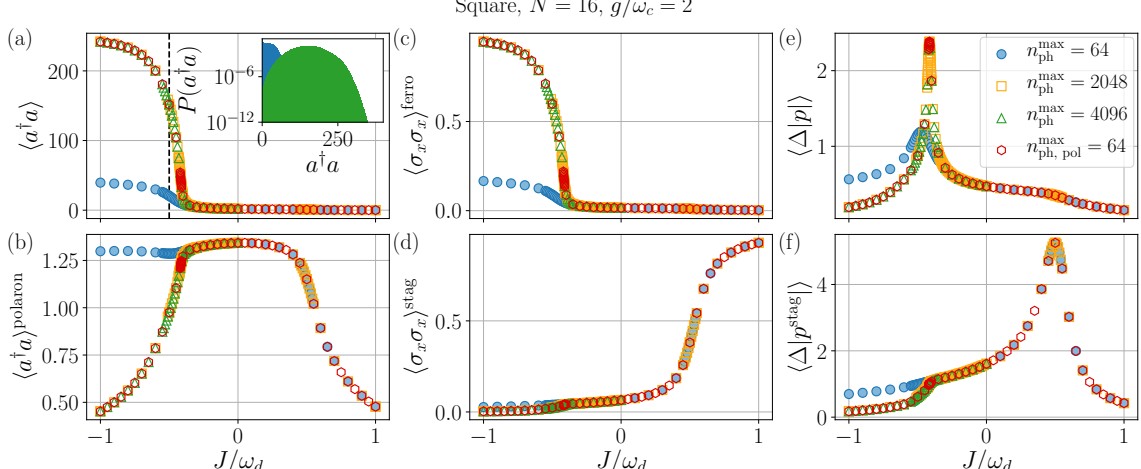

Figure 7: Scaling of observables with the photon number cutoff. Different symbols indicate different photon number cutoffs $n_{\mathrm{ph}}^{\max}$ used in the simulations. The red hexagon shows results of a simulation in the polaron frame, while all other symbols represent simulations in the standard frame. (a) Photon number and (b) polaron photon number, (c-d) order parameter correlations for the (c) ferroelectric, and (d) Néel phase, (e-f) fluctuations of the (e) polarization, (f) staggered polarization. The inset in (a) shows the ground state distribution of the photon number $a^\dagger a$ for $J/\omega_d = -0.5$, as indicated by the vertical dashed line, for $n_{\mathrm{ph}}^{\max} = 64$ (4096) in blue (green) color.

Here, $\mathcal{P}_J$ is the projector onto the classical ground-state manifold. The low-energy eigenvectors of $H_{\mathrm{OBD}}$ yield the states stabilized by the OBD mechanism (in the sense of a first-order degenerate perturbation theory), from which we can compute the observables, as shown in Fig. 5. The advantage of this approach is that, compared to a full simulation of $H_S$, larger system sizes $N$ can be simulated.

# E  Histograms of the three-sublattice order parameter

In this appendix we illustrate the properties of the 3SL order parameter histograms in the complex plain. A (classical) state with sublattice polarizations $\vec{p} = (p_A, p_B, p_C)$ gives a single peak in the histogram according to the axes defined in Fig. 8(a). While this identification is not unique for any $\vec{p}$, the strength of the 3SL ordering $\left|p^{\mathrm{3SL}}\right|$ is given by the distance of the peak from the center, and the angle $\theta$ (in the complex plain) indicates different types of 3SL ordering patterns. In particular, states of type $\vec{p} = (0, 1, -1)$ with two fully, but oppositely polarized, and a non-polarized sublattice (zero net polarization), give peaks at the centers of the hexagonal boundaries of the histogram. The six different peaks correspond to the possible permutations of the three sublattices. States of type $\vec{p} = (1, 1, -1)$, where all sublattices are fully polarized, one of them oppositely to the others (non-zero net polarization) have peaks at the vertices of the hexagonal boundary. The six peaks correspond to the possible permutations of the sublattices and the inversion of the polarization $\vec{p} \to -\vec{p}$.

More generally, patterns of type $\vec{p} = (0, m, -m)$ yield peaks at angles $\theta_l = \pi/6 + l\pi/3$, $l \in \{0, \ldots, 5\}$, with a radius proportional to $m$, as shown in Fig. 8(b). A comparison with the full histograms for the normal and 3SL subradiant regimes [c.f. Fig. 5(a)] shows that the maxima for those phases correspond to such a pattern with maximal $m = 1$, as indicated by the red dots. Furthermore, we want to note that fluctuations of the non-polarized

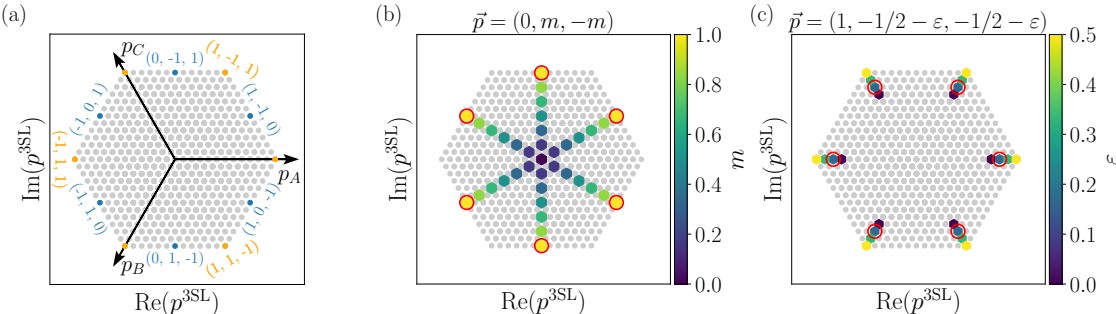

Figure 8: 3SL order parameter histogram in the complex plain. (a) A classical state with sublattice polarizations $\vec{p} = (p_A, p_B, p_C)$ gives a point in the histogram according to the three axes shown in black. The blue dots at the centers of the hexagonal boundary correspond to states of type $\vec{p} = (1, -1, 0)$. The orange dots at the vertices of the hexagonal boundary correspond states of type $\vec{p} = (1, 1, -1)$. (b) The classical ordering pattern $\vec{p} = (0, m, -m)$ (with zero net polarization) yields six peaks along the axis connecting the center to the centers of the boundary faces $[\theta_l = \pi/6 + l\pi/3, \ l \in \{0, \ldots, 5\}]$, where the radius is proportional to the sublattice magnetization $m$. The red circles show the positions of the maxima found in the full histograms of the 3SL normal and 3SL subradiant phases. (c) The classical ordering pattern $\vec{p} = (1, -1/2 - \varepsilon, -1/2 - \varepsilon)$ gives six peaks along the axis connecting the center with the vertices of the outer hexagon $[\theta_l = l\pi/3, \ l \in \{0, \ldots, 5\}]$. The radius of the peaks increases with $\varepsilon$. The red circles show the positions of the maxima found in the full histograms of the 3SL superradiant phase.

sublattice lead to a broadening of the six single peaks parallel to the edges of the hexagonal boundary, as seen for the normal 3SL phase. The strength of these fluctuations can be further used to distinguish the normal and the fluctuation-free subradiant 3SL regimes [c.f. Fig. 5(a)].

On the other hand, patterns of type $\vec{p} = (1, -1/2 - \varepsilon, -1/2 - \varepsilon)$ with a ground state polarization $|S_x| = 2\varepsilon N/3$ have peaks at angles $\theta_l = l\pi/3$, $l \in \{0, \ldots, 5\}$, with a large non-zero radius, which depends on $\varepsilon$, as shown in Fig. 8(c). The red circles show the positions of the maxima found in the full histograms of the 3SL superradiant phase, which has a non-zero net polarization $|S_x| = 2$ ($N = 36$) [c.f. Fig. 5(a)]. We want to note, that all sublattice patterns of type $\vec{p} = (m, -n, -n)$ yield peaks with $\theta_l = l\pi/3$.

## F  Estimating crossover boundaries

In contrast to phase transitions with an abrupt change in the behavior of the order parameter (in the thermodynamic limit), crossovers between two regimes of states with different physical properties show a smooth change (if any) in all of the observables, since the ground state evolves smoothly. Therefore, the crossover region, or a 'boundary' between the two regimes, can typically not be determined uniquely, but depends on the chosen observable and the feature used to estimate the boundary.

To estimate a boundary between the paraelectric regime and the collective subradiant regime, we use maxima in the polaron photon number $\langle a^\dagger a \rangle^{\text{polaron}}$, as shown in Fig. 9(a). In comparison, the standard photon number $\langle a^\dagger a \rangle$ is a bad estimator for the crossover when $J/\omega_c < 0$, where the proximity to the superradiant phase spoils its characteristic features in the narrow collective subradiant regime.

Based on this definition, we observe a shift of the boundary to larger $g/\omega_c$ with increasing

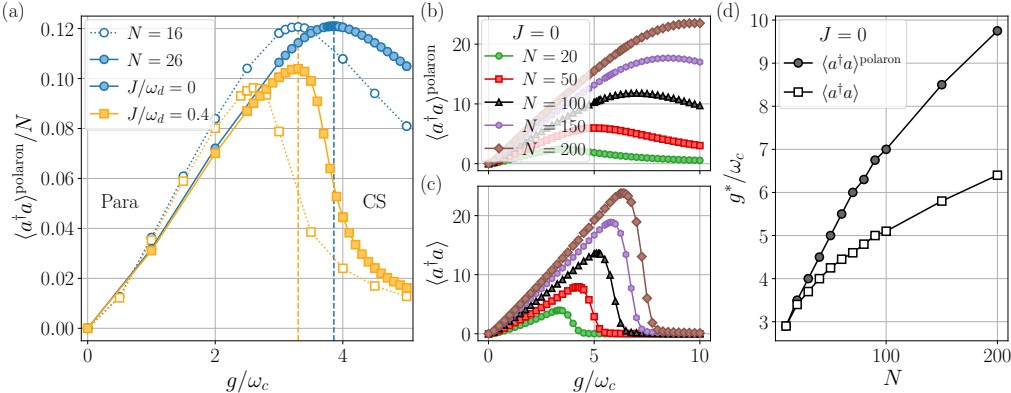

Figure 9: Crossover between paraelectric and collective subradiant regimes. (a) Polaron photon number on an $N = 16$ (open symbols) and an $N = 26$-sites (filled symbols) square lattice for constant values of $J/\omega_d$. We use the peak positions, indicated by the dashed vertical lines for $N = 26$, to estimate the crossover boundary between the paraelectric (Para) and the collective subradiant (CS) regimes [c.f. Fig. 2(a) and Fig. 4(a)]. (b-d) System size dependence of the crossover boundary for $J = 0$. The polaron photon number (b) and normal photon number (c) do not show any trend towards a non-analytic behaviour with system size $N$, which would be a sign for a phase transition. (d) We utilize the maxima in (b, c) to estimate the dependence of the crossover boundary $g^*/\omega_c$ with $N$. Characteristic for a crossover, the boundary depends on the choice of the observable used to define it.

the system size $N$ [see also Fig. 2(a)]. The achievable system sizes in exact diagonalization are too small to make reliable predictions about the fate of the collective subradiant phase in the limit $N \rightarrow \infty$ in general. However, for the non-interacting case $J = 0$ much larger system sizes can be analyzed, as shown in Fig. 9(b-d). Both the polaron and normal photon number increase with system size, but their characteristic shape remains unchanged, so that we do not observe any trend towards a non-analytic behaviour (also for other observables) when increasing $N$, making a phase transition scenario very unlikely. Again, we can use the maxima in these observables to estimate the size dependence of the crossover boundary as shown in Fig. 9(d). While this boundary depends on the choice of observable used to define it, we find a stable collective subradiant phase for all considered system sizes at large enough $g/\omega_c$, consistent with the analysis in [23].

Nevertheless, these numerical results do not provide a conclusive picture about the fate of the collective subradiant phase in the limit $N \rightarrow \infty$. In this respect it is important to emphasize, that this thermodynamic limit is also not properly defined for the single-mode model used in this work, where $H_{\text{cQED}}$ is super-extensive. For finite systems, the single-mode approximation is, however, expected to capture the main results and our analysis can be directly applied to most near-term experiments, where intermediate-scale systems, far away from the thermodynamic limit, will be realized.

Because of the similarity with the evolution from the paraelectric to the collective subradiant regimes, we also expect the evolution from the normal to the subradiant 3SL regime to be described by a crossover instead of a sharp phase boundary. We characterize the regimes by a distinct strength of the polarization fluctuations $\langle \Delta|p| \rangle$, since the polarization fluctuates strongly in the 3SL normal regime and becomes pinned to $S_x = 0$ in the 3SL subradiant regime [see Fig. 4(b)]. We, therefore, define the boundary by a rather sharp drop in $\langle \Delta|p| \rangle$ and estimate its location for constant $J/\omega_d$ by a peak in the negative gradient of $\langle \Delta|p| \rangle$ with respect to the coupling $g/\omega_c$ [see Fig. 10].

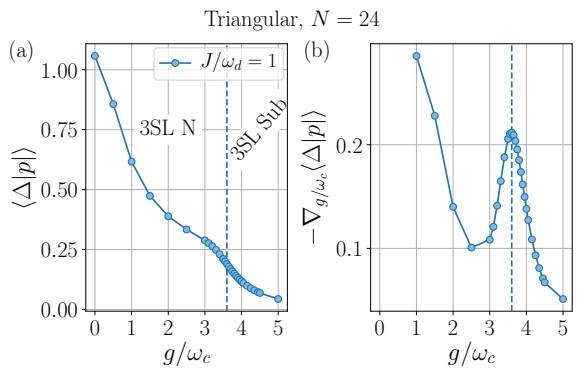

Figure 10: Crossover between the normal 3SL and subradiant 3SL regimes. The polarization fluctuations (a), and their gradient with respect to $g/\omega_c$ (b) are shown for constant $J/\omega_d = 1$. The crossover boundary is estimated by a sharp drop in the fluctuations $\langle\Delta|p|\rangle$, where the polarization distribution becomes strongly pinned to the single value $S_x = 0$. As shown in the right panel, we compute the boundary position by a peak in the negative gradient of $\langle\Delta|p|\rangle$ with respect to the external parameter $g/\omega_c$ (indicated by the dashed vertical line).

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
