# Peer review of "The Vacua of Dipolar Cavity Quantum Electrodynamics"

_SciPost Physics, doi:SciPost Phys. 9, 066 (2020)_

## Round 2 · Referee Report · Anonymous (Referee 1) · 2020-8-3

Strengths

  1. Studies a relevant problem of the interaction between cavity-mediated and short range interactions.

  2. Identifies several novel and unanticipated forms of order - collective subradiant phase, and three sublattice superradiant states.

  3. Shows how some of this physics can be understood by an effective model in the strong matter-light coupling limit.

  4. Provides a good analysis of finite size effects and fluctuations near the phase boundaries.

Weaknesses

In light of SciPost acceptance criterion 6, "Provide (directly in appendices, or via links to external repositories) all reproducibility-enabling resources: explicit details of experimental protocols, datasets and processing methods, processed data and code snippets used to produce figures, etc.;", I note that while this theoretical paper does clearly describe the methods that are used, the authors do not provide a link to a code repository or the direct numerical results of the simulations.

Report

The manuscript by Schuler et al. discusses the phase diagram of a model of dipoles interacting both through a global cavity-mediated interaction, and through local (screened) electrostatic interactions. Such a model has been derived by some of the authors as the multipolar gauge form of effective Hamiltonian in previous work. The key purpose of this paper is to explore the consequences of this Hamiltonian and the nature of its ground state through finite-size exact diagonalization.

The work finds novel phases arise due to the competition of the two interactions, particularly in the context of the frustrated triangular lattice. I believe the results on the appearance of three sublattice superradiant states from the competition of frustrated Ising interactions and long range interactions are significant and could be considered both groundbreaking and as potentially opening up a new area of research. The results are clearly presented, and the manuscript sets this work in context of most other relevant work in this field (see below). For these reasons, I believe the work should be published. There are a few minor issues the authors should consider before publication noted below.

Requested changes

  1. On page 4, when commenting on the importance of including electrostatic screening for a consistent treatment of a cavity, it may be relevant to cite [Andras Vukics and Peter Domokos, Phys. Rev. A 86, 053807 (2012)], which made a similar point.

  2. On page 6, above Eq. 4, there is a statement that the correlations functions are related to second order moments of the phase order parameters, while the equation written in (4) appear to relate them to first order moments of order parameters. This should be clarified.

  3. On page 11, there is a comment about the apparent connection between the current results and supersolidity. In this context, there seems a far more immediate connection to make to experiments on cold atoms in cavities with Raman pumping, where there is competition between long-range cavity mediated interactions and short ranged interactions and hopping. e.g. [Klinder et al., Phys. Rev. Lett. 115, 230403 (2015); Landig et al, Nature 532, 476 (2016)]. While the nature of interactions there differs from the current problem, there may be relevant connections to draw.

  4. On page 11, "has been constraint" should read "has been constrained"

  5. On pages 15/16, in the text of section E and the caption of figure 8, there are references to the states (1,1-1) having peaks at the "edges of the hexagonal boundaries". From the figure, it appears this phrasing is referring to there being peaks at the corners of the hexagonal boundary. If correct, stating these are at the corners would be clearer.

  6. Consider providing open data for the results of the numerical simulation, or access to a code repository to allow reproduction of these simulations.

---

## Round 2 · Referee Report · Anonymous (Referee 2) · 2020-8-16

Strengths

1 - Detailed introduction to models and approximation. 2- Clear definition of the numerical methods used. 3- In-depth analysis of the phases and of their features.

Weaknesses

1- Lack of scaling towards the thermodynamic limit.

Report

In "The Vacua of Dipolar Cavity Quantum Electrodynamics", Schuler and coauthors numerically investigate the ground-state properties of a lattice of $N$ dipoles interacting with a single bosonic mode. The authors take into account both the dipole-dipole interactions (specifically, they consider nearest-neighbors interactions) and the collective interaction of the dipoles with a photonic mode.

The article is extremely interesting from two points of view. On the one hand, as correctly pointed out by the authors, this model describes a cavity QED system in ultrastrong coupling. On the other, it can be seen as a generalization of an Ising model, where local and collective terms also compete with those of a bosonic field.

The article is well-written and deserves publications. I have, however, a major remarks which I think the authors should address. Having answered these two points, I can strongly recommend the publication of this paper.

RMK) The authors provide in very few cases the scaling towards the thermodynamic limit. For a finite-size system, no phase transition can be observed. Since the authors numerically investigate the QED model, i.e., they always consider finite-size models, it is impossible to derive the phase diagram. I would argue that only by studying how the various quantities change in their scaling towards the thermodynamic limit $N \to \infty$, and how they change in proximity of the phase transition, the authors can safely claim the presence of criticality.

I have also these minor remarks: 1) Figures 2(a) and 4(a).
-In the left panels, it is difficult to distinguish between dotted and dashed lines. -In Figure 4(a) the "3SL Superradiant" has the apostrophes in the wrong direction. -The right panels are also very difficult to interpret. I understand what the authors want to do (the vertical axis is for $J$ and the horizontal one is for the quantities in legend) but maybe it can be made slightly more clear by adding the vertical axis labels. That of Figure 2(a) has also two arrows in the vertical axis. 2) In Appendic F, the authors claim "In contrast to phase transitions with an abrupt change in the behavior of the order parameter (in the thermodynamic limit), crossovers between two regimes of states with different physical properties show a smooth change in some of the observables.". This definition is somehow wrong. I would say that it is not the fact that there is "a smooth change in some of the observables" but "a smooth change in all of the observables". Indeed, the phase transition is a nonanalytical change of the ground state. As such, it can be witnessed by the discontinuity of some observables. Thus, finding a discontinuous observable (the order parameter) is sufficient to claim criticality. Vice versa, to claim that there is a crossover one must show that the wave function is continuous, and therefore that all the observables are continuous. Finding a smooth observable does not prove the presence of a crossover.

Moreover, I think that the overall discussion could benefit from addressing these two points (however, they are not necessary for the correct comprehension of the paper): 1) There has been recent controversy on the possibility of using the two-level approximation in the context of ultrastrongly light-matter coupled systems (and some of the authors are part of those article series). To cite some of the papers: * Phys. Rev. A 98, 053819 (2018) * Nat. Commun. 10, 499 (2019) Nat. Phys. 15, 803–808 (2019) arXiv:2005.06499. The authors begin the Section 2 by considering the two level approximation of the anharmonic dipoles. I think the author should justify this approximation given this recent (and still ongoing) debate. 2) Concerning the presence of a crossover between the paraelectric and the collective subradiant phase. In Fig. 2 (a) the boundary seems to move to the right (as discussed by the authors in App. F). In my opinion, however, there are two different questions which need to be answered: 1) Is this really a crossover? 2) What is the position of the crossover (or of the phase transition) for $N \to \infty$? For example, for $J_{i,j}=0$ several different techniques could be used to argue the presence of a crossover (or of a phase transition) between the paraelectric and collective superradiant phase. For example, bosonization of the dipoles via Holstein–Primakoff transformation, or explicitly using the permutational invariance of the spin model, to efficiently diagonalize the Hamiltonian for much larger values of $N$. Another way to understand this feature could be the semiclassical approximation of the bosonized approximation or a Gutzwiller mean-field study of the system.

Requested changes

1- Provide the scaling with $N$ to justify the claims that a crossovers/phase transitions is taking place. 2- Correct the figures. 3- Change the definition of the crossover in Appendix F 4- [FACULTATIVE] Comment on the use of the two-level approximation. 5- [FACULTATIVE] Discuss more in detail the paraelectric to the collective subradiant phase transition for $J=0$, in order to understand the presence of a crossover/phase transition and the position of the boundary.

---

## Round 2 · Referee Report · Anonymous (Referee 3) · 2020-9-7

Strengths

1)Interesting model and results, rich phase diagram (role of frustration, order-by disorder due to cavity, collective subradiant superradiant phases, etc..)

2) Solid numerical work

3)well written

Weaknesses

1)physics in the photonic sector could be more discussed (see changes)

2)few key points in the text could be better explained (see changes)

3) figures are a bit hard to read (too much information), captions do not always help

Report

This manuscript by Schuler et al discusses the ground-state phase diagram of a two-dimensional model of interacting dipoles (two-level systems) coupled to a cavity mode, relevant for CavityQED platforms in the ultra-strong coupling limit. The model can be seen as intermediate between Ising and Dicke models, and in this sense shares some similarities with the Rabi-Hubbard model of CavityQED.

The results, obtained with exact diagonalization (after truncating the photon Hilbert space) reveal a very rich phase diagram, in particular on the triangular lattice which is in my opinion one of the highlight of this work.

I think the results are relevant and worth to be published in SciPost Physics. At the same time I have a number of comments on the manuscript that I would kindly request the authors to consider. I append them below.

Requested changes

1)The authors discuss the broken symmetry phases in terms of dipolar order, but do not touch upon whether the photon field also breaks the symmetry and becomes coherent, i.e. \neq 0 in the thermodynamic limit, as I would expect. Could the authors clarify this point? 2) One could then ask whether photonic and dipolar order always come hand by hand (as often in Dicke-like models) or whether one could have exotic situations where in one sector the symmetry is broken but in the other the order is still fluctuating. Is this scenario conceivable in the present model, for example in presence of frustration, i.e. on the triangular lattice? This would be pretty neat I think. 3)Following up on the points above, I notice that in terms of numerics the focus is mainly on photon number, which is ok but not really the relevant photonic "order parameter". Clearly for finite size systems the average photon field would be always zero but I wonder whether there could be interesting signatures of criticality in the low frequency retarded photon Green's function, which should be easy to compute with ED. In principle this should diverge a zero frequency at the critical point. 4)Few points in the manuscript could be expanded/clarified. For example the discussion on pag6 on "collective subradiant", it is not clear the connection with the g->infty limit. Similarly, the discussion below Eq.9 could be expanded and the connection between the effective Hamiltonian and the emergence of a super-radiant phase in the top right corner of the phase diagram. 5)Figures are a bit dense and the caption, although long and detailed does not always help. I could not find a discussion of the central panel in figures 2 and 4. In particular Figures 2 and 4 could be perhaps split in multiple figures?

---

## Round 3 · Referee Report · Anonymous (Referee 2) · 2020-10-3

Report

The author addressed the main criticisms and raised points in a satisfactory way. The present version of the article is clear, and the possible shortcomings of finite size analysis have been clarified throughout the paper.

As such, I can recommend the publication of the paper as it is.

---

## Round 3 · Author Response

We thank all referees for their detailed analysis and reports of our manuscript and appreciate their very positive feedback. We provide detailed replies to the individual reports below:

Report 1:
We appreciate pointing out SciPost Acceptance criterion 6 to provide reproducibility-enabling resources. For that, we have now uploaded all data presented in this paper and code snippets to reproduce the figures to an online repository at https://doi.org/10.5281/zenodo.4018821.

Regarding the other comments, we have adopted the manuscript accordingly (see list of changes below).

Report 2:
(1) We agree with the referee that for finite systems there are no phase transitions and no criticality in a strict sense. However, even though the range of available system sizes is limited in the current numerical analysis, it still provides strong evidence that the presented phase diagrams qualitatively describe the phase diagrams for large systems:

(i) All phases are well characterized by a specific set of observables, which are large throughout the phase and decrease rather abruptly at the boundaries.

(ii) The distinctly ordered phases we observe can be described by symmetry-breaking states in the thermodynamic limit and many of them connect to well-established phases in the transverse field Ising model limit.

(iii) Transitions between the phases are accompanied by sharp peaks in the fluctuations of the respective order parameters.   

While small shifts of the phase boundaries may still occur, none of our findings indicates qualitative changes in the phase diagrams that might occur for larger N. Therefore, we believe that speaking of a phase diagram is well-justified in the present context. Note that we are also very careful not to make any definite statements about the nature of the phase transitions or their critical behavior and clearly point out that such questions cannot be answered rigorously by our finite-size numerical calculations. A direct finite-size scaling analysis is currently far out of reach for any exact analysis of this model.

Finally, let us emphasize that HcQED describes the coupling of all dipoles to a single cavity mode with fixed properties, in particular, a fixed interaction region. Simply increasing the system size N does not represent a well-defined thermodynamic limit, while a rescaling of the coupling constant, g → g/\sqrt{N}, would render the dipole-field coupling non-perturbative. Therefore, the finite-size phase diagrams discussed in the present manuscript are representative for the practically relevant scenarios, where small or mesoscopic ensembles of dipoles are coupled a field mode localized within a tiny mode volume. 

We have modified the discussion in section 3 to better convey this message.

(2) We have changed the appearance of the Figures 2 and 4 and added a separate panel for the right plots to make them easier to read.

(3) We thank the referee for pointing out the unclear definition of a crossover in Appendix F, in which we wanted to point out that some observables might not show any characteristic change at all, and that the crossover boundary depends on the observable which is considered. Indeed no observable is non-analytic at a crossover and we have rephrased this definition in Appendix F.

(4) Hamiltonian (1) has been derived in Ref. [2] and in the follow-up paper Phys. Rev. A 98, 053819 (Ref. [47] in the new version of our manuscript) the choice of the dipole gauge for performing a systematic two-level approximation has been justified analytically and numerically in great detail. We are aware that the issue about the validity of the two-level approximation in different gauges has since then attracted quite some attention. However, we would like to emphasize that this debate is mainly about alternative derivations and not about the validity of the original results. Indeed, when applied to the same physical setting all the different predictions reduce to the dipole- and Coulomb gauge Hamiltonians presented in [2].

We don't think that the discussion about the choice of gauge is particularly relevant for the current analysis, but since the reader might wonder about this point we added a brief statement together with some of the main references on this topic below Eq. (1).

(5) Our analysis of the evolution from the paraelectric to the collective subradiant state shows indeed no trend towards a non-analytical behaviour in any of the observables we have computed. Since also both of these regimes do not break any symmetries, we expect this evolution to be better described by a smooth crossover, instead of a sharp phase transition.

In the non-interacting limit Ji,j=0 larger systems can be diagonalized exactly and we have added data for much larger systems up to N=200 dipoles in the latest version of the manuscript [see Fig. 9 (b-d)]. These simulations still do not show any signs of non-analytic behaviour and further support the scenario of a smooth crossover, but do not yet provide a conclusive picture about the behaviour of the crossover boundary in the limit N→∞. 

The crossover from the paraelectric to the subradiant phase remains an interesting open problem since it occurs in a regime, where both the validity of the Holstein-Primakoff approximation (valid for small couplings) and the effective spin model HS [Eq.(6) in the latest version of the manuscript] (valid for very large couplings) break down. Also, the subradiant phase is a highly entangled state which is not captured by a mean-field ansatz or a semiclassical approximation. So far we are not aware of any analytic approach which is able to reproduce the numerical findings in the crossover region for large N.

Finally, we want to again emphasize that the limit N→∞ does not represent a well-defined thermodynamic limit in our model and that our analysis is already representative for the practically relevant scenarios, in particular with the now presented large system size results for Ji,j=0.

We have also adopted the presentation of the collective subradiant phase in the manuscript to better illustrate that the crossover scenario is strongly supported by our results.

Report 3:
(1) As discussed in the manuscript, on finite-size systems symmetries cannot be broken spontaneously. Therefore, the order parameter < a > but also the polarization < p > are always strictly zero in the ground state. However, second-order moments of the order parameters (such as < aa >, < a^\dagger a > or < p^2 > and < \sigma_x \sigma_x >), become non-zero in the corresponding regimes and are typically used as "finite-size order parameters”. From our finite-size analysis, we thus expect < a > and also < p > to be non-zero in the ferroelectric phase when the models Z_2 symmetry is eventually spontaneously broken in the thermodynamic limit. 

Furthermore, it is important to note, that the polaron transformation illustrates, that superradiant phases with non-zero < p >=< S_x > directly yield coherent states for the photons with < a > = g/\omega_c < S_x >.

We have rephrased section 3 of the manuscript to further emphasize and clarify this important point.

(2) Since the Z_2 symmetry operator of the model is a combination of dipole and photon operators, photonic and dipolar order are always related to each other, which is also obvious from the polaron transformation.
However, unlike in ferroelectric phases where the coherent photon state with < a > = g/\omega_c < S_x > is directly related to the polarization of the dipoles, we illustrate in our manuscript in detail that more complex scenarios are possible. In particular, the anti-aligned Néel phase, which breaks the Z_2 symmetry in a non-trivial way, has < S_x > = 0 (in the thermodynamic limit) and thus features a vanishing photon field. So, in this case the symmetry can be considered unbroken for the photon sector.
Even more, the 3SL normal phase on the triangular lattice features strong photon number fluctuations beyond a coherent state due to the inherent fluctuations of the dipoles, while the latter are ordered in a non-trivial way.

(3) Because of the intimate relation between properties of the photonic and the dipolar sector in the ordered phases, we mainly focus on a discussion of dipolar observables in the manuscript to obtain the phase boundaries. An analysis of the photonic sector, beyond what is already shown, will give identical phase transitions in the ordered phases.
The retarded photon Green’s function is, however, not a very good quantity to estimate phase transitions on finite-size systems. Since the energy gaps at critical points will always be finite it will not diverge at zero frequency, and a detailed finite-size scaling analysis would be necessary to see traces of this divergence.

Since such an analysis would not yield new properties of the ground state phase diagrams, it goes far beyond the scope of this manuscript.

(4) The collective subradiant phase exists in the limit J=0 and g/\omega_c>>1, as illustrated in Fig. 2 of our manuscript and discussed in Ref. [23]. In this limit, one can derive the effective model H_S from which the ground state |\psi_{cs}> with the discussed features can be derived.

We have modified sections 3 and 5 to make these points clearer in our presentation.

(5) We have changed the appearance of figures 2 and 4 and have moved the central plot into a new panel (b). This should make the figures easier to read and, in particular, the captions better understandable. We are, however, convinced that the splitting the figures, as suggested, would reduce their readability, since the data among the individual panels is related.

---

## Round 3 · List of Changes

• Added references: Phys. Rev. A 86, 053807 (2012); Phys. Rev. Lett. 115, 230403 (2015); Nature 532, 476 (2016); Phys. Rev. A 98, 053819 (2018); Nat. Commun. 10, 499 (2019); Nat. Phys. 15, 803–808 (2019); arXiv:2005.06499;
  • We have fixed Eq. 4 which now correctly relates to the second-order moments.
  • Rephrased section 3 to present our results in a clearer way.
  • Rephrased section 5 to improve readability.
  • Modified figures 2 and 4 to make them easier to read.
  • Added an additional panel to figure 3.
  • Added panels to figure 9 with results for larger systems at J=0.
  • Rephrased the definition of a crossover in Appendix F and extended the discussion.
  • Changed "edges" -> "vertices" in Appendix E.
  • Fixed typos.

---

## Editorial Decision

published